# A billion years arms-race between viruses, virophages, and eukaryotes

Jose Gabriel Nino Barreat, Aris Katzourakis*

Department of Biology, University of Oxford, Oxford, United Kingdom

**Abstract** Bamfordviruses are arguably the most diverse group of viruses infecting eukaryotes. They include the Nucleocytoplasmic Large DNA viruses (NCLDVs), virophages, adenoviruses, *Mavericks* and Polinton-like viruses. Two main hypotheses for their origins have been proposed: the 'nuclear-escape' and 'virophage-first' hypotheses. The nuclear-escape hypothesis proposes an endogenous, *Maverick*-like ancestor which escaped from the nucleus and gave rise to adenoviruses and NCLDVs. In contrast, the virophage-first hypothesis proposes that NCLDVs coevolved with protovirophages; *Mavericks* then evolved from virophages that became endogenous, with adenoviruses escaping from the nucleus at a later stage. Here, we test the predictions made by both models and consider alternative evolutionary scenarios. We use a data set of the four core virion proteins sampled across the diversity of the lineage, together with Bayesian and maximum-likelihood hypothesis-testing methods, and estimate rooted phylogenies. We find strong evidence that adenoviruses and NCLDVs are not sister groups, and that *Mavericks* and Mavirus acquired the rve-integrase independently. We also found strong support for a monophyletic group of virophages (family *Lavidaviridae*) and a most likely root placed between virophages and the other lineages. Our observations support alternatives to the nuclear-escape scenario and a billion years evolutionary arms-race between virophages and NCLDVs.

**\*For correspondence:**
aris.katzourakis@biology.ox.ac.uk

**Competing interest:** The authors declare that no competing interests exist.

## eLife assessment

The **important** study by Barreat and Katzourakis examines the evolutionary history of eukaryotic viruses (and related mobile elements) in the *Bamfordvirae* kingdom, and evaluates potential alternative scenarios regarding the origin of different lineages in this highly diverse kingdom. Through **convincing** phylogenetic analyses, the authors propose a new evolutionary model for the origin of this kingdom where their last common ancestor is inferred to have been an exogenous, non-virophage DNA virus with a small genome. This work advances our understanding of the deep evolutionary history of viruses, the interaction between viruses and the first eukaryotes, and the diversification of viral lineages.

## Introduction

Viruses in the kingdom *Bamfordvirae* comprise one of the most diverse groups in terms of their genome complexity, ecology and morphology. These dsDNA viruses include the largest viruses characterised to date (*Legendre et al., 2014*; *Scheid, 2016*), virophages which are virus parasites of other viruses (*Fischer and Suttle, 2011*; *La Scola et al., 2008*), endogenous viruses that colonise the genomes of eukaryotes (*Kapitonov and Jurka, 2006*; *Pritham et al., 2007*), and a diverse set of viruses that have been identified from metagenomic data (*Bellas and Sommaruga, 2021*; *Moniruzzaman et al., 2020*; *Paez-Espino et al., 2019*; *Roux et al., 2017*; *Schulz et al., 2020*; *Yutin et al., 2015*). More formally, these viruses are classified into the Nucleocytoplasmic Large DNA Viruses (NCLDVs) which comprise families in the phylum *Nucleocytoviricota* (*Aylward et al., 2021*), the virophage parasites of NCLDVs (family *Lavidaviridae*; *Fischer, 2021*), adenoviruses (family *Adenoviridae*) which infect vertebrates

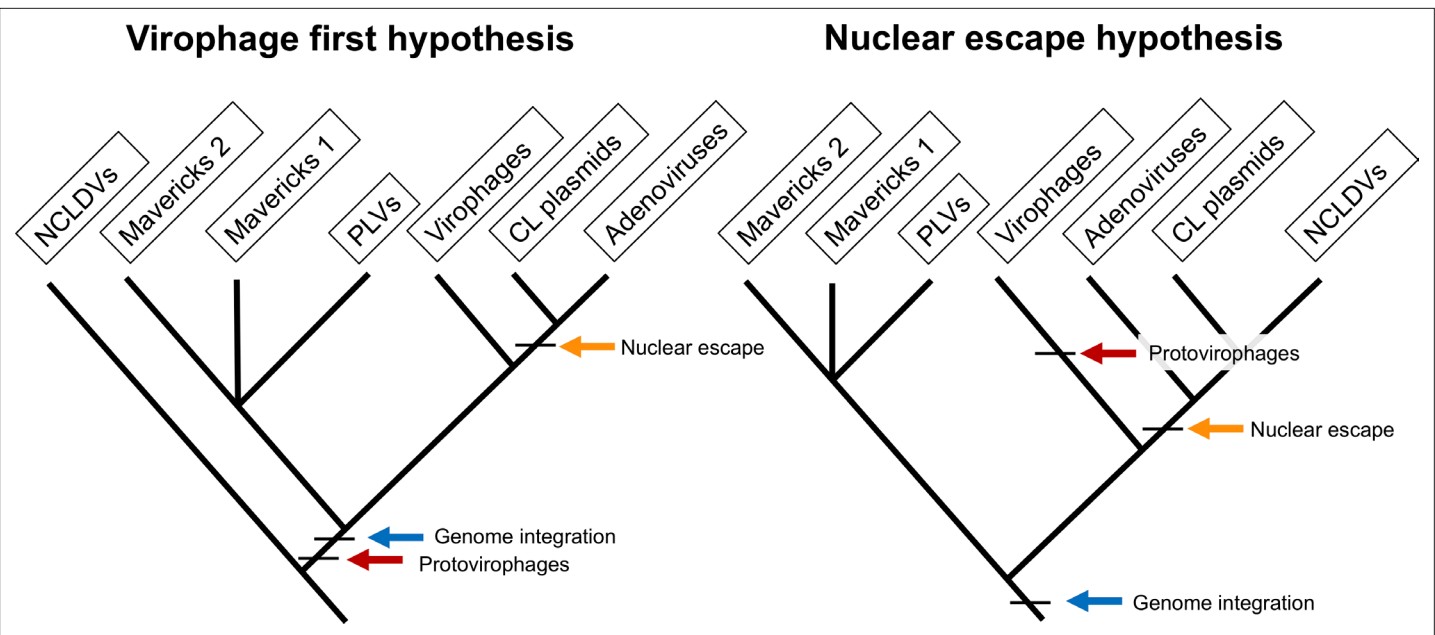

**Figure 1.** The two main hypotheses for the origin of virophages and NCLDVs. In the virophage first hypothesis, NCLDVs diverge early with its sister lineage evolving into protovirophages. In the nuclear escape hypothesis, NCLDVs descend from endogenous elements (encoding an integrase) that became exogenous; virophages then evolved to become their parasites.

(*Harrach et al., 2019*), the *Maverick/Polinton* endogenous viruses that colonise the genomes of eukaryotes (*Kapitonov and Jurka, 2006*; *Pritham et al., 2007*), Polinton-like viruses (PLVs) which are abundant in aquatic metagenomes (*Bellas and Sommaruga, 2021*; *Yutin et al., 2015*), and capsidless elements such as transpovirons (*Desnues et al., 2012*), mitochondrial and cytoplasmic linear plasmids (*Krupovic and Koonin, 2015*; *Meinhardt et al., 1997*). The wide taxonomic distribution of NCLDV and *Maverick* hosts, which comprise all major eukaryotic lineages (*Kapitonov and Jurka, 2006*; *Pritham et al., 2007*; *Schulz et al., 2020*), seem to point to an ancient origin of these viral groups.

Despite their diversity, the elements share an ancestral gene module used for capsid morphogenesis, which is the feature that distinguishes viruses from other mobile genetic elements (*Koonin et al., 2021*). The ancestral module is formed by double and single jelly-roll capsid proteins, a family C5 (adenoviral-like) protease and a family FtsK/HerA DNA packaging ATPase (*Krupovic et al., 2016c*; *Yutin et al., 2015*). In fact, the jelly-roll capsid proteins and DNA packaging ATPase, also occur in phages from the families *Turriviridae*, *Tectiviridae*, *Corticoviridae,* and *Autolykiviridae*, which suggests a common viral ancestor for eukaryotic bamfordviruses and their prokaryotic virus relatives (*Koonin et al., 2020*; *Woo et al., 2021*). These observations are at odds with the suggestion that NCLDVs originated by reductive evolution, possibly from a fourth domain of cellular life (*Colson et al., 2018*; *Legendre et al., 2012*; *Patil and Kondabagil, 2021*).

Currently, two hypotheses have been proposed for the origin of the eukaryotic *Bamfordvirae*: the 'nuclear-escape' and 'virophage-first' scenarios (*Figure 1*). According to the nuclear-escape scenario, the elements in this lineage evolved from an endogenous virus in the nucleus of an early eukaryote, a *Maverick*-like ancestor (*Koonin and Krupovic, 2017*; *Krupovic and Koonin, 2015*; *Krupovic and Koonin, 2016a*). Adenoviruses, cytoplasmic linear plasmids and NCLDVs evolved after their ancestor escaped from the nucleus (*Koonin et al., 2015a*; *Krupovic and Koonin, 2015*). Therefore, the nuclear-escape hypothesis predicts that adenoviruses form a clade with NCLDVs and cytoplasmic linear plasmids, and that the proteins shared by cytoplasmic linear plasmids and NCLDVs have a single common origin (*Koonin and Krupovic, 2017*; *Krupovic and Koonin, 2015*). The nuclear-escape hypothesis is based on phylogenetic trees of the protein-primed DNA polymerase B, which show a paraphyletic group of *Mavericks* at the base of the other lineages (*Koonin and Krupovic, 2017*; *Krupovic et al., 2016c*; *Krupovic and Koonin, 2015*). However, the trees are missing NCLDVs which encode a non-homologous nucleotide-primed DNA polymerase, and could therefore not be included in the

analyses, and only show the position for Mavirus-like virophages which encode the protein-primed DNA polymerase.

The virophage-first scenario is based on the close similarity in gene content between the Mavirus virophage and *Mavericks*, and the ability of Mavirus to integrate into eukaryotic genomes (*Fischer and Hackl, 2016*; *Fischer and Suttle, 2011*). Integrated Mavirus virophages reactivate upon giant virus infection and confer protection against *Cafeteria roenbergensis virus* in the flagellate *Cafeteria burkhardae* (*Fischer and Hackl, 2016*). The virophage-first scenario therefore suggests that an ancestral virophage evolved shortly after the origin of NCLDVs, and was able to parasitise NCLDVs by virtue of their shared promoter and poly-A sequences (*Campbell et al., 2017*; *Fischer and Suttle, 2011*). One of these lineages of ancestral virophages would have gained an integrase and endogenised into eukaryotic hosts providing an immune defence system, which gave rise to *Mavericks* and other elements (*Fischer and Suttle, 2011*; *Katzourakis and Aswad, 2014*). A nuclear-escape would have still occurred for adenoviruses and cytoplasmic linear plasmids, but not NCLDVs, which would belong to a different clade suggesting the immediate ancestor of NCLDVs was an exogenous virus.

Phylogenetics can be used to test competing evolutionary models of virus origins in a rigorous statistical framework. Each tree represents an evolutionary hypothesis which is inferred from matrices of aligned homologous characters with different states (the data). In molecular phylogenetics, nucleotide or amino acid alignments are used as the data to obtain the topology of a tree (and also rates or branch lengths). In statistical terms, the data matrices are used to calculate the tree likelihood (probability of the data given the parameters) in both maximum-likelihood and Bayesian frameworks (*Schmidt and von Haeseler, 2009*; *Nascimento et al., 2017*). In maximum likelihood, the preferred tree is the one with the parameter values that maximise the tree likelihood (*Schmidt and von Haeseler, 2009*). In

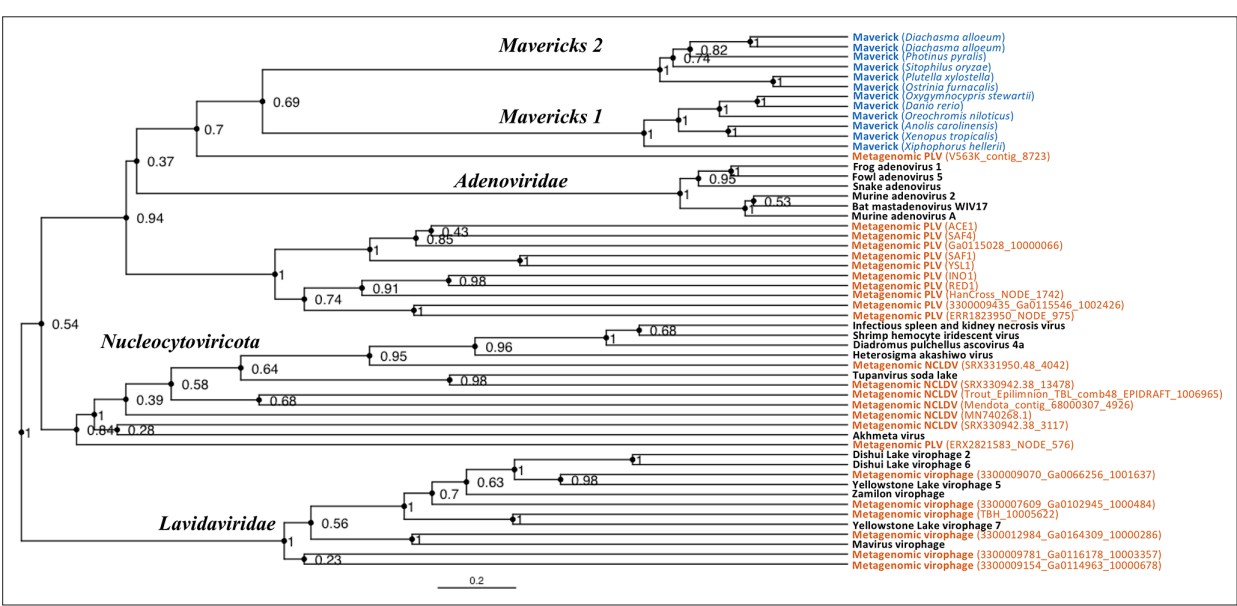

**Figure 2.** Rooted Bayesian maximum clade credibility tree of the major lineages of eukaryotic viruses in the kingdom *Bamfordvirae*. The tree is based on the concatenated alignment of 4 core proteins involved in virion morphogenesis (major and minor capsid proteins, ATPase and protease). Tree computed in BEAST 2 (*Bouckaert et al., 2019*) using a relaxed molecular clock and 140 million generations (relative burn-in of 25%). Black: reference viral genomes, Blue: endogenous elements, Orange: metagenomic sequences.

The online version of this article includes the following figure supplement(s) for figure 2:

**Figure supplement 1.** Bayesian maximum credibility tree of the major capsid protein inferred with a relaxed molecular clock.

**Figure supplement 2.** Bayesian maximum credibility tree of the minor capsid protein inferred with a relaxed molecular clock.

**Figure supplement 3.** Bayesian maximum credibility tree of the ATPase inferred with a relaxed molecular clock.

**Figure supplement 4.** Bayesian maximum credibility tree of the protease inferred with a relaxed molecular clock.

**Figure supplement 5.** Bayesian maximum credibility tree of the ATPase inferred with a relaxed molecular clock, and using a monophyletic constraint on virophages.

**Figure supplement 6.** Bayesian maximum credibility tree of the protease inferred with a relaxed molecular clock, and using a monophyletic constraint on virophages.

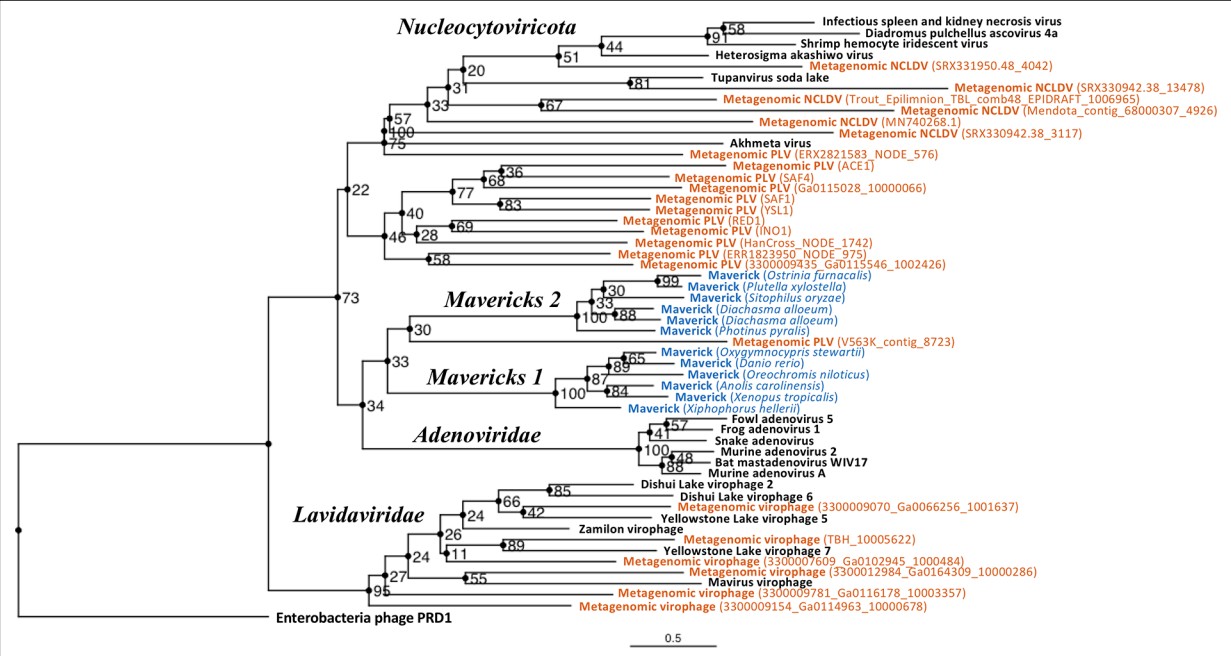

**Figure 3.** Outgroup-rooted maximum likelihood tree of the major lineages of eukaryotic viruses in the kingdom *Bamfordvirae*. The tree is based on the concatenated alignment of 4 core proteins involved in virion morphogenesis (major and minor capsid proteins, ATPase and protease). *Enterobacteria phage PRD1* (*Tectiviridae*) was used as the outgroup for rooting. Tree computed in RAxML-NG with 200 random starting trees and 2200 bootstraps (*Kozlov et al., 2019*). Black: reference viral genomes, Blue: endogenous elements, Orange: metagenomic sequences.

The online version of this article includes the following figure supplement(s) for figure 3:

**Figure supplement 1.** Maximum-likelihood unrooted phylogenetic trees of the transcriptional homologues encoded in cytoplasmic linear plasmids.

Bayesian inference, a posterior probability distribution for the parameters is estimated from their prior distributions and the tree likelihood (probability of the parameters given the data) (*Nascimento et al., 2017*). Together, these approaches can be used to assess the plausibility of competing evolutionary models that make different predictions about tree topologies and the position of the root.

Understanding the diversification of these viruses remains a major open question in virus evolution. To address this problem, we have analysed a data set of virus representatives sampled across the diversity of the eukaryotic bamfordvirus lineage and focused on the 4 core virion proteins: major and minor capsid proteins, protease and ATPase. We use explicit hypothesis-testing methods and estimate the position of the root in our phylogenies, which allowed us to infer the direction of evolution followed by the proteins in the morphogenetic module. Our results suggest a new model for the evolutionary origins of these viruses, which is consistent with an exogenous, non-virophage ancestor.

## Results

The rooted Bayesian and maximum-likelihood trees estimated from the concatenated data had the best overall support and shared important topological similarities (*Figures 2 and 3*). In the Bayesian tree, adenoviruses were placed in a clade with *Mavericks* and most PLVs with a high posterior probability (0.94, *Figure 2*), and to the exclusion of NCLDVs. This arrangement was also observed in the maximum likelihood tree topology shown in *Figure 3*, although with a low bootstrap support (34%). The virophages (family *Lavidaviridae*) emerged as a strongly supported monophyletic group on both trees (posterior probability = 1 in the Bayesian analysis, 95% bootstrap support in the maximum-likelihood analysis), while Polinton-like viruses were clearly polyphyletic.

The position of the root inferred both through the Bayesian relaxed-clock analysis and by outgroup rooting with the *Enterobacteria phage PRD1* (*Tectiviridae*), was between virophages and all the other viral lineages. In the Bayesian analysis, the root was placed on this position in 54% of trees (*Figure 2*), while in the maximum-likelihood analysis the clade of (Adenoviruses + *Mavericks* + PLVs+NCLDVs), that is to the exclusion of virophages, received a bootstrap support of 73% (*Figure 3*). We also

**Table 1.** Comparison between the nuclear-escape and alternative maximum-likelihood models based on the Akaike information criterion (AIC).

The log-likelihoods for the models were obtained from the best maximum-likelihood tree consistent with each hypothesis found in RAxML-NG (*Kozlov et al., 2019*). Results are shown for the concatenated data set of four core proteins (ATPase, protease, major capsid, and minor capsid) and for the protease, major and minor capsid proteins, respectively. The AIC was size-corrected given that the alignment (sample) size was small relative to the number of free parameters, that is n/k<40 (*Posada and Buckley, 2004*; *Symonds and Moussalli, 2011*). The best model is highlighted in boldface. The 'alternatives' refer to non 'nuclear-escape' scenarios, that is, models which are not consistent with the predictions made by the nuclear-escape.

| Model | Characters | Log-likelihood | AIC* | AICc† | ΔAICc‡ | Weight§ |
|---|---|---|---|---|---|---|
| Nuclear-escape (M$_0$) | Concatenated | –41,618.19 | 83,616.37 | 83,864.09 | 11.41 | 0.003 |
| Alternatives (M$_1$) | Concatenated | –41,612.48 | 83,604.96 | 83,852.68 | 0.00 | 0.997 |
| Nuclear-escape (M$_0$) | Protease | –6,323.233 | 12,898.47 | 10,898.22 | 11.37 | 0.003 |
| Alternatives (M$_1$) | Protease | –6,317.550 | 12,887.10 | 10,886.85 | 0.00 | 0.997 |
| Nuclear-escape (M$_0$) | Major capsid | –16,305.52 | 32,861.04 | 34,173.54 | 1.90 | 0.279 |
| Alternatives (M$_1$) | Major capsid | –16,304.57 | 32,859.14 | 34,171.64 | 0.00 | 0.721 |
| Nuclear-escape (M$_0$) | Minor capsid | –10,234.56 | 20,719.11 | 17,855.48 | 5.62 | 0.057 |
| Alternatives (M$_1$) | Minor capsid | –10,231.75 | 20,713.50 | 17,849.86 | 0.00 | 0.943 |

$^{*}AIC_i = -2 \cdot logL_i + 2 \cdot K$.

$^{†}AICc_i = AIC_i + \frac{2K(K+1)}{(n-K-1)}$.

$^{‡}\Delta AICc_i = AICc_i - min\left(AICc_i, AICc_j\right)$.

$^{§}Weight_i = \frac{e^{-\Delta AICc_i/2}}{\sum_i^j e^{-\Delta AICc_i/2}}$.

examined alternative placements of the root sampled during the Bayesian Markov-chain Monte Carlo (MCMC) to assess the frequency of other hypotheses (*Supplementary file 2*). Rooting on the clade formed by NCLDVs + PLV BS_539 (ERX2821583_NODE_576) was observed in 27.4% of trees. The root positions at the next highest frequencies were on the branch of PLV BS_539 (5.6%), NCLDVs (5.3%) and PLV BS_539 + NCLDVs + virophages (5%). Roots on other branches were observed at frequencies lower than 1% (*Supplementary file 2*).

The monophyly of virophages and the root on this branch were also observed in the single-protein trees. The major and minor capsid single-protein trees recovered a monophyletic *Lavidaviridae* (0.97 and 0.52 posterior probability), and also inferred the root to be placed between virophages and the other viruses (29% and 28%, *Figure 2—figure supplement 1*, *Figure 2—figure supplement 2*). Although the root was observed at a different position for the ATPase and protease trees; these analyses also failed to recover a monophyletic family of virophages (*Figure 2—figure supplement 3*, *Figure 2—figure supplement 4*). However, when virophages were constrained to be monophyletic the root was placed again between virophages and the other lineages, and thus in agreement with our findings using the concatenated data, major and minor capsid proteins (*Figure 2—figure supplement 5*, *Figure 2—figure supplement 6*).

The Bayesian and maximum-likelihood tests of topology were also consistent with each other. We used the Akaike information criterion (AIC) to compare the plausibility of the maximum-likelihood topologies consistent with the nuclear-escape and alternative scenarios. All the comparisons preferred a non-sister grouping of adenoviruses and NCLDVs (*Table 1*). Indeed, the Akaike weights, which can be interpreted as the normalised relative likelihood of the models given the data (*Posada and Buckley, 2004*), strongly favoured the alternative scenario over the nuclear-escape hypothesis (Concatenated data = 99.7%, Protease = 99.7%, Major capsid protein = 72%, Minor capsid protein = 94.3%). The approach using the posterior model odds, calculated following the method of *Bergsten et al., 2013*, also favoured the alternative scenarios in all of the data sets: concatenated and single-protein (*Table 2*). This was strongest for the concatenated data (posterior model odds, H$_0$/H$_1$=0.000536), but

**Table 2.** Posterior model odds of the nuclear-escape and alternative hypotheses using concatenated and single-protein data sets.

Tree topologies consistent with each hypothesis were filtered and counted from a Bayesian MCMC following the method of *Bergsten et al., 2013*. All ratios favour the alternative scenarios to the nuclear-escape hypothesis. The best model is highlighted in boldface. The 'alternatives' refer to non-'nuclear-escape' scenarios, that is, models which are not consistent with the predictions made by the nuclear-escape.

| Model | Characters | MCMC tree frequency | Posterior model odds $P(M_0 \mid X)/P(M_1 \mid X)$ |
|---|---|---|---|
| Nuclear-escape ($M_0$) | Concatenated | 15/28,001 (0.0536%) | |
| **Alternatives ($M_1$)** | **Concatenated** | **27,986/28,001 (99.946%)** | $5.36 \cdot 10^{-4} < 1$ |
| Nuclear-escape ($M_0$) | Protease | 333/200,001 (0.166%) | |
| **Alternatives ($M_1$)** | **Protease** | **199,668/200,001 (99.998%)** | $1.66 \cdot 10^{-3} < 1$ |
| Nuclear-escape ($M_0$) | Major capsid | 20,605/200,001 (10.302%) | |
| **Alternatives ($M_1$)** | **Major capsid** | **179,396/200,001 (89.697%)** | $1.15 \cdot 10^{-1} < 1$ |
| Nuclear-escape ($M_0$) | Minor capsid | 128/200,001 (0.0639%) | |
| **Alternatives ($M_1$)** | **Minor capsid** | **199,873/200,001 (99.936%)** | $6.39 \cdot 10^{-4} < 1$ |

the same pattern held for the single-protein analyses (posterior model odds = {Protease: 0.00166, Major capsid: 0.11485, Minor capsid: 0.000639}). The Bayesian stepping-stone analysis on the concatenated data also strongly supported the alternative scenario (NCLDVs and adenoviruses are not sister groups), over the nuclear-escape hypothesis (NCLDVs and adenoviruses are sister groups; *Table 3*).

As an independent test of the predictions made by the nuclear-escape hypothesis, we analysed the origin of the transcriptional proteins shared between NCLDVs and cytoplasmic linear plasmids. According to the nuclear-escape hypothesis, these genes were acquired by the most recent common ancestor of cytoplasmic linear plasmids and NCLDVs, and should thus have a single origin. However, our analyses indicate that these proteins were acquired independently by cytoplasmic linear plasmids and NCLDVs. The maximum-likelihood trees for the Rbp2 subunit of the DNA-directed RNA polymerase, helicase and mRNA-capping proteins all suggest they have independent origins in NCLDVs and cytoplasmic linear plasmids (*Figure 3—figure supplement 1*). The homologues specific to cytoplasmic linear plasmids clustered with those of eukaryotes and their distribution agrees with a monophyletic origin in this group. Interestingly, the phylogenetic patterns for NCLDVs were consistent with a single-capture (monophyletic origin) for the mRNA-capping enzyme and the helicase, while the distribution of the Rpb2 was consistent with multiple-captures/exchanges (polyphyletic origin).

**Table 3.** Bayesian stepping-stone analysis of the nuclear-escape and alternative hypotheses.

Each scenario was run on the concatenated data set in MrBayes 3 (*Ronquist and Huelsenbeck, 2003*) for 20 million generations (average standard deviation of split frequencies <0.01). The Bayes factor strongly rejects a sister relationship between adenoviruses and NCLDVs (nuclear-escape hypothesis). The best model is highlighted in boldface. The 'alternatives' refer to non 'nuclear-escape' scenarios, that is, models which are not consistent with the predictions made by the nuclear-escape.

| Model | Likelihood of best state (cold chain) | Log-Marginal-likelihood (ln) | Mean Log-marginal-likelihood (ln) | Bayes factor $P(X \mid M_0)/P(X \mid M_1)$ |
|---|---|---|---|---|
| Nuclear-escape ($M_0$) | Run 1: −36,318.81 | Run 1: −36,376.42 | −36,353.44 | |
| | Run 2: −36,318.81 | Run 2: −36,352.75 | | $3.5 \times 10^{-94} \ll 1^{*}$ |
| **Alternatives ($M_1$)** | **Run 1: −35,965.90** | **Run 1: −36,137.56** | **−36,138.25** | |
| | **Run 2: −35,981.46** | **Run 2: −36,450.19** | | |

*Strong support against $M_0$.

## Discussion

To gain a better understanding of the evolutionary history of the bamfordviruses of eukaryotes, we analysed a set of four core virion morphogenesis proteins (major and minor capsids, DNA-packaging ATPase and protease) sampled across the diversity of the lineage and using an explicit hypothesis-testing framework. In addition, we analysed the origins of three proteins shared by cytoplasmic linear plasmids and NCLDVs. We found strong evidence against the nuclear-escape hypothesis. We also found support for a position of the root between virophages (family *Lavidaviridae*) and the other viral lineages. This position of the root suggests a new evolutionary model for the origin of these viral lineages. However, we recognise that some alternative root positions were also observed at lower frequencies in the Bayesian analyses, which could be interpreted as 'virophage-first'-like scenarios.

We have shown that adenoviruses and NCLDVs are not sister groups as proposed by the nuclear-escape hypothesis (*Koonin and Krupovic, 2017*; *Krupovic and Koonin, 2015*). Instead, adenoviruses belong to a clade with *Mavericks*, to the exclusion of NCLDVs. The phylogenies of the proteins shared by cytoplasmic linear plasmids and NCLDVs also disagree with a nuclear escape. Indeed, phylogenies of the protein-primed DNA polymerase place cytoplasmic linear plasmids as the sister-group to adenoviruses (*Krupovic et al., 2016c*), implying that adenoviruses, cytoplasmic linear plasmids and NCLDVs form a clade. If a single nuclear-escape had occurred we would expect the Rbp2, helicase and mRNA capping enzymes shared by cytoplasmic linear plasmids and NCLDVs to have a monophyletic origin (since they descend from the same 'escaped' ancestor). However, their distribution agrees with a more recent origin in cytoplasmic linear plasmids. These observations agree with the more restricted taxonomic distribution of cytoplasmic linear plasmids, which are known only to infect fungi in the Order Saccharomycetales (Division Ascomycota; *Meinhardt et al., 1997*), in contrast to the taxonomically diverse eukaryotic hosts of NCLDVs.

Virophages emerged as a highly supported monophyletic group in our analyses. This supports the validity of the family *Lavidaviridae* which has been based on shared gene contents of virophages and their parasitic (or commensal) lifestyles (*Fischer, 2021*; *Krupovic et al., 2016b*). By using two independent rooting methods, relaxed-molecular clocks and outgroup rooting, we found that the root was placed most confidently between virophages and the other lineages; making them the most basal lineage of eukaryotic bamfordviruses. The basal position of virophages has also been found in an independent work that looked at phylogenies based on the ATPase and a concatenation of the ATPase and the major capsid protein (*Woo et al., 2021*). This basal position of virophages and their divergence prior to the origin of NCLDVs, suggests that the protovirophages were not parasites of other viruses. Instead, they would have become parasites of other viruses at a later point once the most recent common ancestor of NCLDVs had evolved (see Ideas and speculation).

The tree topologies that we estimated (*Figure 2* and *Figure 3*), also suggest that the most recent common ancestor of *Mavericks*, adenoviruses and PLVs was not a virophage, which is at odds with a virophage-first scenario. Indeed, the basal position of virophages is inconsistent with the virophage-first hypothesis which proposes that protovirophages coevolved with NCLDVs, acquired an integrase and then gave rise to *Mavericks* and other elements in the lineage (*Campbell et al., 2017*; *Fischer and Suttle, 2011*). However, we found that the second-highest frequency root sampled in the Bayesian MCMC was on the branch of PLV BS_539 + NCLDVs (frequency = 27.4%). Rooting on this branch would be consistent with a 'virophage-first'-like scenario, where the lineage of NCLDVs and associated viruses evolved early and in parallel with protovirophages. Therefore, we cannot definitely rule out a 'virophage-first'-like scenario. In contrast, root placements on the branch leading to *Mavericks* and consistent with nuclear-escape were observed in 0% of the trees, which seems to rule out this possibility (*Supplementary file 2*).

The relative timing of events suggests that the most recent common ancestor of eukaryotic bamfordviruses existed more than a billion years ago. There is general agreement based on paleontological and molecular evidence that the Last Eukaryotic Common Ancestor (LECA) existed at least 1 billion years ago (*Porter, 2020*). Analyses of genetic exchanges between the DNA-dependent RNA polymerases of NCLDVs and eukaryotes have determined that the two super-clades of NCLDVs (PAM and MAPI) already existed before the origin of LECA (*Guglielmini et al., 2019*). Moreover, discovery and phylogenetic analysis of actin and actin-related proteins encoded in NCLDVs, also suggest that early gene transfers occurred between these viruses and proto-eukaryotes before the emergence of LECA (*Da Cunha et al., 2022*). The basal placement that we observed for virophages, or alternatively,

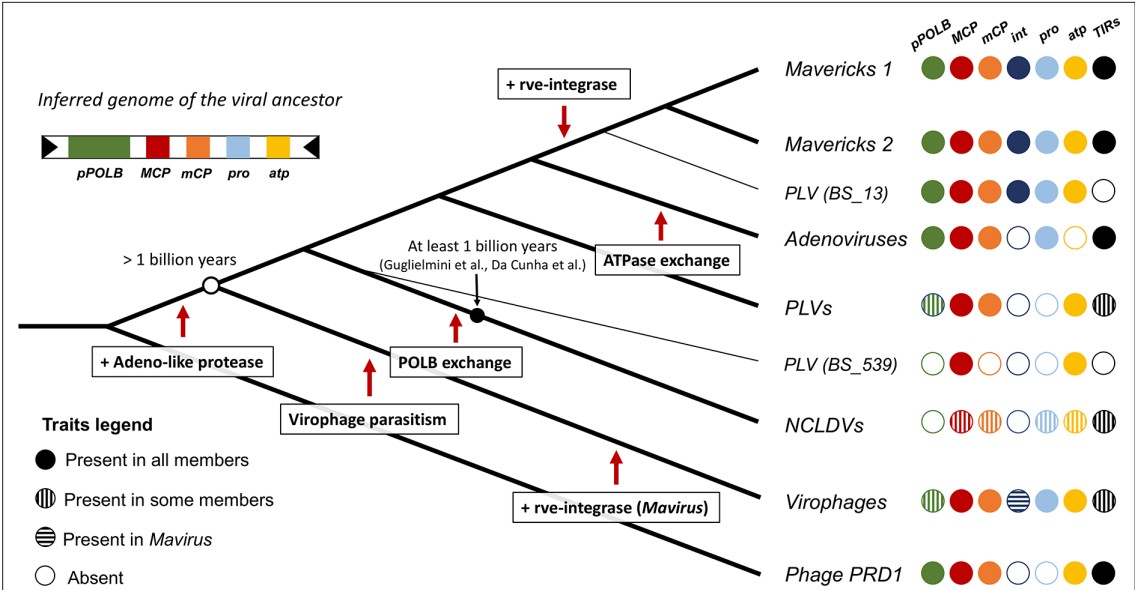

**Figure 4.** Evolutionary model for the origin of the major lineages of eukaryotic viruses in the kingdom *Bamfordvirae*. The viral ancestor is inferred to have been an exogenous virus, while the rve-integrase was captured independently by the clade of Mavericks + Polinton like virus BS_13 and Mavirus, possibly by horizontal gene transfer. Virophages evolved from an autonomous virus that became specialised to parasitise the ancestor of NCLDVs. The vertical cross-hatching indicates that the trait is found in some but not all members of the group. Acronyms refer to genes and genomic features present in the viral genomes: (pPOLB) (protein-primed DNA polymerase B) , (MCP) (major capsid protein), (mCP) (minor capsid protein), (int) (rve-type integrase), (pro) (adenoviral-like protease), (atp) (FtsK/HerA DNA packaging ATPase), (TIRs) (terminal inverted repeats).

their early origin after the divergence of the NCLDV lineage ('virophage-first'-like scenario), suggest that virophages evolved after the origin of the first NCLDVs during the diversification of the early eukaryotes. These observations, together with the existence of various phage relatives that infect bacteria and archaea, point to a very early origin of the kingdom *Bamfordvirae*, perhaps extending to the initial stages of cellular life. Studies that consider the time-dependent rate phenomenon in reconstructing viral evolution may be able to shed light on these issues (*Aiewsakun and Katzourakis, 2016*; *Ghafari et al., 2021*).

## Ideas and speculation

Our analyses favour a new model for the evolution of the major lineages of eukaryotic bamfordviruses; we call this model the 'exogenous, non-virophage scenario'. The model is presented in *Figure 4*, where we have mapped shared character states onto a diagram of the concatenated Bayesian phylogeny (which has the best overall support). According to this idea, the most recent common ancestor of the eukaryotic bamfordviruses was a small exogenous dsDNA virus with a linear genome, flanked by inverted repeats. The virus had an autonomous (i.e. non-virophage) lifestyle. It used a protein-primed DNA polymerase B for replication and its morphogenetic module was formed by the major and minor jelly-roll capsid proteins, an adenoviral-like protease and the FtsK/HerA family DNA-packaging ATPase. The gain of the adenoviral-like protease differentiated this eukaryotic lineage from their phage relatives (*Krupovic and Koonin, 2015*). The first divergence from this ancestor led to the emergence of the protovirophages, and another lineage that would give rise to *Mavericks*, adenoviruses, PLVs, and NCLDVs (*Figure 4*).

The second divergence would have given rise to the lineage from which NCLDVs evolved. The most recent common ancestor of NCLDVs is believed to have already had a complex genome (~40 genes), which involved the gain of numerous genes from their eukaryotic hosts, other viruses and bacteria (*Iyer et al., 2006*; *Koonin and Yutin, 2010*; *Yutin and Koonin, 2012*). Once this virus evolved the capacity to exploit significant cell resources, there would have been a window of opportunity for protovirophages to evolve into specialised parasites of NCLDVs, occupying this new ecological niche. The close match between virophage and NCLDV regulatory sequences, which underlie the capacity of virophages to parasitise NCLDVs, may have evolved by parallel evolutionary changes on the sequences

of their shared virus ancestor. It is plausible that functional regulatory sequences in virophages could evolve *de novo* by a few changes in the ancestral promoters/terminators. For example, functional *lac* operon promoters have been evolved successfully even from random sequences in the presence of lactose in *E. coli* (**Yona et al., 2018**). A deletion mutant in the ORF 8 gene of the Guarani virophage expanded its host range to previously non-permissive giant viruses, so the acquisition and muta-tion of proteins involved in virus-virophage interactions may also be critical for virophage adaptation (**Mougari et al., 2020**). Interestingly, the exchange of the protein-primed DNA polymerase B by a nucleotide-primed DNA polymerase B (**Mönttinen et al., 2021**; **Yutin and Koonin, 2012**), may have been an early counter-measure of NCLDVs to decrease the parasitic burden imposed by virophages. The acquisition of this new polymerase, may be in part responsible for the considerable increases in genome size that we see in NCLDVs relative to other eukaryotic bamfordviruses. Therefore, we can hypothesise that the large genomes of NCLDVs, which is one of their most distinctive features, could have evolved as a result of their ancient interaction with virophages.

The distribution of the rve-integrase on the phylogeny suggests it was acquired independently on two separate occasions. Indeed, the rve-integrase is a universal feature of Mavericks + PLV BS_13, while it does not seem to be present in any of their closest relatives. Therefore, the rve-integrase seems to be a unique derived feature of the clade of Mavericks + PLV BS_13. Mavirus also encodes a rve-integrase but it appears to be the only virophage with this gene (no other hits found in a blastp search of the nr database using the labels 'Lavidaviridae' + 'unclassified *Lavidaviridae*'). Since Mavirus is firmly nested within the phylogenies for the morphogenetic module, it seems likely that this is a unique derived feature of Mavirus. The most parsimonious explanation for these observations is that the integrase gene was acquired on two separate occasions and that the common ancestor of Mavirus and *Mavericks* was an exogenous virus without an integrase (# character state changes = 2 gains). This is in line with a previous proposal that Mavirus evolved by recombination of an ancestral virophage with a *Maverick* (**Yutin et al., 2013**).

## Concluding remarks

Our findings strengthen the view of a viral origin for NCLDVs and other eukaryotic elements in the kingdom *Bamfordvirae*. We have decisively shown that adenoviruses and NCLDVs are not sister groups, and that adenoviruses, virophages, PLVs and NCLDVs did not escape from the nucleus. The rve-integrase appears to have been captured independently by Mavericks +PLV BS_13 on the one hand, and by Mavirus on the other. These observations support a new evolutionary model that is different from the nuclear-escape and virophage-first scenarios. This new model proposes that the ancestor of the eukaryotic viruses in the kingdom *Bamfordvirae* was an exogenous, non-virophage DNA virus with a small genome. The lifestyle of virophages would have evolved at a later stage as these became specialised parasites of the ancestral NCLDVs. Darwin's closing words in the *Origin of Species* are well-suited for this group of viruses: 'from so simple a beginning endless forms most beautiful and most wonderful have been, and are being, evolved' (**Darwin, 1859**).

# Materials and methods

We used maximum-likelihood and Bayesian hypothesis-testing methods to compare the plausibility of the nuclear-escape versus alternative evolutionary scenarios. We focused on the 4 core virion proteins shared by viruses in this lineage: major and minor capsid proteins, ATPase and protease. Rooted phylogenies were inferred from the data using a relaxed molecular-clock or an outgroup rooting method. We then assessed whether adenoviruses and NCLDVs formed a monophyletic group as predicted by the nuclear-escape scenario. Finally, we studied the origin of the proteins shared by cytoplasmic linear plasmids and NCLDVs. A more detailed description of methods is presented below.

## Selection of viral sequences

We first compiled a data set of viruses belonging to the families *Phycodnaviridae*, *Mimiviridae*, *Asco-viridae*, *Iridoviridae*, *Marseilleviridae*, *Asfarviridae*, *Poxviridae*, *Adenoviridae,* and *Lavidaviridae* that were obtained from the NCBI using accession numbers listed in the International Committee for the Taxonomy of Viruses (ICTV) master species list 2020 .v1. Previous analyses have shown that *Maver-icks/Polintons* fall into two major groups based on the protein-primed DNA polymerase: the group I

*Mavericks* (that include elements from vertebrates) and the group II *Mavericks* (which include elements present in insects) (*Kapitonov and Jurka, 2006*). Group I *Mavericks* can be distinguished from group II elements by the presence of a ~140 aa insertion similar to a bacterial 'very-short-patch' (VSR) repair endonuclease (*Kapitonov and Jurka, 2006*). To make sure both groups were represented in our analyses, we included the vertebrate *Mavericks* reported in *Barreat and Katzourakis, 2021*, and added 12 new intact *Maverick* elements that we discovered in the genomes of insects and which lack the VSR insertion (*Supplementary file 1*).

We also included the sequences for Polinton-like viruses, virophages and NCLDVs that have been reported in the metagenomic works of *Bellas and Sommaruga, 2021*, *Moniruzzaman et al., 2020*, *Schulz et al., 2020*, *Paez-Espino et al., 2019*, *Roux et al., 2017* and *Yutin et al., 2015*. Additional genomes for medusavirus (*Yoshikawa et al., 2019*), cedratvirus (*Jeudy et al., 2020*), tupanviruses (*Abrahão et al., 2018*), pandoraviruses (*Antwerpen et al., 2015*; *Philippe et al., 2013*; *Legendre et al., 2018*), pithovirus (*Legendre et al., 2014*), mollivirus (*Legendre et al., 2015*), insectomime virus (*Boughalmi et al., 2013*), tokyovirus (*Takemura, 2016*), kaumoebavirus (*Bajrai et al., 2016*), faustoviruses (*Benamar et al., 2016*), pacmanvirus, orpheovirus and the Dishui lake virophages (*Sayers et al., 2021*), were also included. In total, the data set consisted of 9,222 virus genomes.

## Protein prediction and subsampling approach

We devised a subsampling approach to arrive at a representative subset of sequences to carry out the phylogenetic analyses and tests of topology. First, open-reading frames for all sequences were predicted with getorf in EMBOSS (*Rice et al., 2000*). Next, we used a set of Hidden Markov Models (hmmbuild/hmmsearch, HMMER version 3.3.2 using the default parameters) for each of the 4 core proteins involved in virion morphogenesis (major and minor capsid proteins, ATPase and protease) to extract homologous proteins in 9 groups: genomic/metagenomic virophages, genomic/metagenomic NCLDVs, metagenomic PLVs, PLVs described in *Yutin et al., 2015*, adenoviruses, and *Mavericks 1* and *2* lineages. We used the major capsid protein to make multiple sequence alignments in MAFFT (*Katoh et al., 2002*) and construct approximate phylogenies for each group in FastTree 2 (*Price et al., 2010*). A total of 2803 major capsid proteins could be identified in the genome data, which comprised: adenoviruses (71 sequences), *Mavericks 1/2* (28 sequences), NCLDVs NCBI/metagenomic (2,031 sequences), PLVs Yutin/metagenomic (327 sequences), and virophages NCBI/metagenomic (346 sequences). The trees were used to subsample 6 representatives from each of the 9 groups maintaining the maximal diversity in Treemmer (*Menardo et al., 2018*). The final data set contained representatives of 54 taxa (*Supplementary file 1*).

## Protein multiple sequence alignment

Using the list of representative taxa for the major capsid proteins, we retrieved the corresponding sequences for the ATPase, protease and minor capsid proteins in each genome. We confirmed that each prediction corresponded to its homology group using HHpred (*Soding et al., 2005*). Sequences were aligned in MAFFT (*Katoh et al., 2002*) and trimmed using trimAl (*Capella-Gutiérrez et al., 2009*). We then built a concatenated alignment using a custom script written in Python (*van Rossum et al., 2009*).

## Bayesian phylogenetic inference

We constructed single-protein and concatenated rooted trees in BEAST 2 (*Bouckaert et al., 2019*). Best models for amino acid substitution were selected for each partition in ModelTest-NG (*Darriba et al., 2020*). The model LG +G + I (with 4 categories for the gamma rate heterogeneity) was used for the major capsid, protease and ATPase, while the model LG +G (with 4 categories for the gamma rate heterogeneity) was used for the minor capsid protein. We imposed monophyletic constraints at the level of well-established viral clades/families: NCLDVs, *Adenoviridae*, vertebrate *Mavericks* and invertebrate *Mavericks* (however, the monophyly between both groups of *Mavericks* was not constrained). The monophyly of virophages and Polinton-like virophages were not constrained. We conducted a Bayesian MCMC with a chain length of 140 million generations for the concatenated data set and 200 million generations for the single proteins (sampling every 5,000th generation and using relaxed molecular clocks). Convergence was assessed by inspecting the runs for good-mixing and stationarity,

and ensuring that Effective Sample Sizes were >200 for all parameters in Tracer (*Rambaut et al., 2018*).

## Maximum-likelihood phylogenetic inference

We estimated a maximum-likelihood tree in RAxML-NG (*Kozlov et al., 2019*) using the concatenated data set and the same monophyletic constraints on well-established groups as described above. Proteins of the *Enterobacteria phage PRD1* (ATPase, major, and minor capsid) were included in the alignment for outgroup rooting. We chose the best substitution models in ModelTest-NG (*Darriba et al., 2020*). For the protease and ATPase, we used model LG +G + I+F (with 4 categories for the gamma rate heterogeneity), and for the major and minor capsid proteins we used model LG +G + F (with four categories for the gamma rate heterogeneity). The analysis was started from 200 random starting trees and run with 2200 bootstrap replicates.

## Akaike information criterion

To compare the plausibility of the maximum-likelihood trees consistent with each hypothesis, we inferred the best supported phylogenies in RAxML-NG and then performed model-selection based on the Akaike information criterion (*Kozlov et al., 2019*; *Posada and Buckley, 2004*). Specifically, we compared a constrained model consistent with nuclear-escape where adenoviruses and NCLDVs are sister groups, to a model where the sister relationship between adenoviruses and NCLDVs was not constrained (alternative model). We included a partitioned analysis of the concatenated data set and three core proteins individually (protease, major and minor capsid), using the amino acid substitution models described in the previous section. The analyses were run with 200 random starting trees and we kept the estimate for the log-likelihood of the best tree model for the calculation of the AICs. The AIC for each model was size-corrected given that the alignment (sample) sizes were small compared to the number of free parameters in each model, or $n/k<40$ (*Posada and Buckley, 2004*; *Symonds and Moussalli, 2011*). For the comparison, we calculated the Akaike weights which can be interpreted as the normalised relative likelihood of the model given the data (*Posada and Buckley, 2004*), or as an analogue of the probability that a model is the best approximating model given the data within a set of candidates (*Symonds and Moussalli, 2011*). The ATPase was not included in the single-protein analyses since adenoviruses encode a non-homologous ABC family ATPase (*Burroughs et al., 2007*), and therefore it cannot be used to test their evolutionary relationships to other bamfordviruses.

## Posterior model odds

The tree topologies obtained in the Bayesian MCMCs for the single proteins and the concatenated data set, were loaded and filtered in PAUP (*Swofford, 1998*). Following the posterior model odds method of *Bergsten et al., 2013*, we counted the number of trees which contained a sister grouping of adenoviruses and NCLDVs (nuclear-escape hypothesis) and compared them to the number of trees inconsistent with this grouping (alternative hypotheses). The posterior model odds were calculated as the ratio of these two quantities (*Bergsten et al., 2013*). The ATPase was again excluded from the single-protein analysis given that adenoviruses could not be included.

## Marginal likelihood and Bayes factors

We estimated the marginal likelihood of each hypothesis using stepping-stone sampling. We ran the analysis on the concatenated data set, for 20 million generations in MrBayes 3 (*Ronquist and Huelsenbeck, 2003*). For both analyses, we used the monophyletic constraints described above. In the marginal calculation for the nuclear-escape hypothesis, we used a topological constraint on the sister grouping of adenoviruses and NCLDVs, while we imposed a negative topological constraint on the monophyly of adenoviruses and NCLDVs for the alternatives (adenoviruses and NCLDVs cannot be sister groups). We calculated the Bayes factors as the number *e* to the power of the difference in log-marginal likelihoods, and compared the resulting value to the table of *Kass and Raftery, 1995*.

## Cytoplasmic linear plasmids

Cytoplasmic linear plasmids have lost the ancestral module of genes involved in formation of the capsid, but they have gained three genes encoding a mRNA capping-enzyme, a helicase and the Rpb2 subunit of the DNA-dependent RNA-polymerase II (*Koonin et al., 2015b*; *Krupovic and*

*Koonin, 2015*). We used the proteins in cytoplasmic linear plasmids (*Supplementary file 3*), to search for homologous sequences in the non-redundant protein database using blastp (*Altschul et al., 1990*) and restricting searches to 'Eukaryota (taxid:2759)' and 'Viruses (taxid:10239)'. The identity of significant matches (evalue <1e-10) was confirmed with HHpred (*Soding et al., 2005*), and in the case of eukaryotes, only sequences mapping to chromosome assemblies were used. Virus, eukaryotic and plasmid homologues were aligned in MAFFT (*Katoh et al., 2002*) and conserved blocks recovered from trimAl (*Capella-Gutiérrez et al., 2009*). We chose the best models for protein evolution in ModelTest-NG (*Darriba et al., 2020*) and ran a maximum-likelihood tree search in RAxML-NG (*Kozlov et al., 2019*) with 1000 non-parametric bootstrap replicates for each protein separately.

## Acknowledgements

We wish to thank Peter Simmonds and Alexander Suh for their critical reading and comments on the manuscript which served to improve this work. We also thank the reviewers for their recommendations and feedback. This work was supported by a doctoral scholarship (Dr. Jose Gregorio Hernandez Award) to JGNB made by the National Academy of Medicine of Venezuela and Pembroke College, Oxford.

## Additional information

### Funding

| Funder | Grant reference number | Author |
| --- | --- | --- |
| National Academy of Medicine of Venezuela | Dr. Jose Gregorio Hernandez Award | Jose Gabriel Nino Barreat |
| Pembroke College Oxford | Dr. Jose Gregorio Hernandez Award | Jose Gabriel Nino Barreat |

The funders had no role in study design, data collection and interpretation, or the decision to submit the work for publication.

### Author contributions

Jose Gabriel Nino Barreat, Conceptualization, Data curation, Software, Formal analysis, Validation, Investigation, Visualization, Methodology, Writing – original draft, Writing – review and editing; Aris Katzourakis, Conceptualization, Supervision, Methodology, Project administration, Writing – review and editing

### Author ORCIDs

Jose Gabriel Nino Barreat http://orcid.org/0000-0002-4589-9473
Aris Katzourakis http://orcid.org/0000-0003-3328-6204

Public Review: https://doi.org/10.7554/eLife.86617.3.sa1
Author Response: https://doi.org/10.7554/eLife.86617.3.sa2

## Additional files

### Supplementary files

• Supplementary file 1. Viral species included in the final multiple sequence alignments (54 taxa) with their accession numbers and source reference.

• Supplementary file 2. Distribution of root positions calculated from the MCMC posterior tree sample. The best supported position of the root was on the branch leading to virophages (53.9%), followed by NCLDVs and metagenomic PLV BS539 (27.4%). Other root positions received <6% support. The frequencies of trees with a certain position of the root were estimated by filtering different topologies in PAUP. Number of generations = 140 million.

• Supplementary file 3. Cytoplasmic linear plasmids used for querying the databases in search for

protein homologues.
- MDAR checklist

## Data availability

The Hidden Markov Models used to find homologues of the four core proteins (n = 38), and concatenated and single-protein alignments used for phylogenetic inference have been deposited in Figshare. The scripts used for sequence concatenation and for the AIC analyses are available on GitHub (copy archived at *Barreat, 2023*).

The following dataset was generated:

| Author(s) | Year | Dataset title | Dataset URL | Database and Identifier |
|---|---|---|---|---|
| Barreat JGN, Katzourakis A | 2023 | Multiple sequence alignments and HMMs of the four core virion proteins of eukaryotic bamfordviruses | https://doi.org/10.6084/m9.figshare.19576117 | figshare, 10.6084/m9.figshare.19576117 |

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
