## [Editor Report · eLife assessment]

The **important** study by Barreat and Katzourakis examines the evolutionary history of eukaryotic viruses (and related mobile elements) in the *Bamfordvirae* kingdom, and evaluates potential alternative scenarios regarding the origin of different lineages in this highly diverse kingdom. Through **convincing** phylogenetic analyses, the authors propose a new evolutionary model for the origin of this kingdom where their last common ancestor is inferred to have been an exogenous, non-virophage DNA virus with a small genome. This work advances our understanding of the deep evolutionary history of viruses, the interaction between viruses and the first eukaryotes, and the diversification of viral lineages.

---

## [Referee Report · Public Review]

Barreat and Katzourakis analyze the evolutionary history of eukaryotic viruses (and related mobile elements) in the Bamfordvirae kingdom, and discuss potential scenarios regarding the origin of different viral taxa in this group. This version of their manuscript now includes a larger number of sequences to better represent diversity in these viral groups, and explored new evolutionary scenarios, including a "virophage-first" hypothesis now presented as the one best supported by phylogenetic analyses. The authors also present compelling analyses suggesting that the "nuclear escape" hypothesis in which these different viral groups separately "escaped" from nuclear (integrated) elements is not consistent with the current genomic and phylogenetic information available.

This work is thus an important step in our collective understanding of the ancient evolutionary history of eukaryotic viruses, and more generally of the constraints and main drivers of virus evolution.

---

## [Author Response]

The following is the authors' response to the original reviews.

1. l. 80: "evolved from a fourth domain of cellular life": I am worried a little bit about putting together what I believe are too distinct hypothesis: (i) NCLDV deriving from a complex (ancestral) cellular life form (possibly proto-eukaryotic) by reductive evolution, and (ii) NCLDV forming or deriving from a fourth domain of cellular life. To clarify for non-expert reader, I would suggest rephrasing as "evolved reductive evolution, possibly from a fourth domain of cellular life...".

Following the reviewer’s recommendation, we have clarified the sentence by writing: “These observations are at odds with the suggestion that NCLDVs originated by reductive evolution, possibly from a fourth domain of cellular life (Colson et al., 2018; Legendre et al., 2012; Patil and Kondabagil, 2021).”.

2. l. 187-198: Please provide more information on which tool (with version number and parameter) was used to search genomes for MCPs. When I downloaded the HMM model and the faa file for the MCP from the figshare repository and tried to match the two, only a small number (4) of the MCP sequences actually matched the MCP HMM model with significant e-value, but I am not sure why? (for reference, I was using hmmsearch 3.3.2, default parameters)

We used HMMER version 3.3.2 using the default parameters (hmmbuild and hmmsearch algorithms). We now include this information in the relevant section of the Methods: “Next, we constructed a set of Hidden Markov Models (HMMER version 3.3.2, hmmbuild/hmmsearch using the default parameters) for each of the 4 core proteins involved in virion morphogenesis”.

We were able to reproduce the reviewer’s observation that the Major capsid curated HMM model returns 4 significant hits when used on the Major capsid multiple alignment file provided in FigShare (significant matches: 1. maverick2_NW_021681489.1_105940131438, 2. ncbincldv_NC_011335.1, 3. ncbincldv_NC_038553.1, 4. yutin_PLVACE1). This curated HMM model was one of the models used for searching homologous protein sequences and was built from a preliminary multiple sequence alignment comprising a different set of taxa (N. taxa = 48). In contrast, the multiple sequence alignment provided in Figshare is the final multiple sequence alignment of major capsid proteins that was used in phylogenetic analyses (N. taxa = 54). Therefore, we should not expect an exact match between the two files.

We have updated the Figshare repository with a compressed file containing all the HMMs used for searching protein homologues (n = 38), which can be validated on hmmsearch on the European Bioinformatic Institute’s website (https://www.ebi.ac.uk/Tools/hmmer/search/hmmsearch). A separate compressed file contains the final multiple sequence alignments that were used in phylogenetic inference and hypothesis testing.

3. Figure 4: The acronyms should be explained in the legend (pPOLB, MCP, mCP, pro, atp, int, TIRs, etc)

We now provide an explanation of the acronyms used for the traits matrix on Figure 4: “Acronyms refer to genes and genomic features present in the viral genomes: *pPOLB* (protein-primed DNA polymerase B), *MCP* (major capsid protein), *mCP* (minor capsid protein), *int* (rve-type integrase), *pro* (adenoviral-like protease), *atp* (FtsK/HerA DNA packaging ATPase), *TIRs* (terminal inverted repeats).”

4. Figure 4: I believe that "TIRs" should be "Present in some members" for the virophages, based on https://doi.org/10.1186/s13062-015-0054-9? Interestingly, this group is typically the one that branches the deepest within virophages, which would be consistent with TIRs being an ancestral trait of the Maveriviricetes class (formerly Lavidaviridae family).

As suggested, we updated the terminal inverted repeats (TIRs) trait for virophages to “Present in some members” to account for the Rumen virophages described by Yutin, Kapitonov and Koonin (2015, doi: 10.1186/s13062-015-0054-9).

**Additional changes:**

1. Figure 1 has been updated and now shows a polytomy between Mavericks 1/2 and PLVs. This reflects more closely the conceptual framework for our analyses since the specific branching of these groups was not specified in the phylogenetic models.

2. We have added an Acknowledgements section to the end of the manuscript:

**Acknowledgements**

We wish to thank Peter Simmonds and Alexander Suh for their critical reading and comments on the manuscript, which served to improve this work. We also thank the reviewers for their recommendations and feedback. This work was supported by a doctoral scholarship (Dr. Jose Gregorio Hernandez Award) to JGNB made by the National Academy of Medicine of Venezuela and Pembroke College, Oxford.